# The Possibilities of Immunotherapy for Children with Primary Immunodeficiencies Associated with Cancers

**DOI:** 10.3390/biom10081112

**Published:** 2020-07-28

**Authors:** Frederic Baleydier, Fanette Bernard, Marc Ansari

**Affiliations:** 1 Department for Women, Children and Adolescents, Paediatric Haemato-Oncology unit, Geneva University Hospital, CH-1211 Geneva, Switzerland; fanette.bernard@hcuge.ch (F.B.); marc.ansari@hcuge.ch (M.A.); 2 CANSEARCH research laboratory, Medical Faculty, Geneva University, 1205 Geneva, Switzerland

**Keywords:** immunotherapies, cancers, primary immunodeficiencies

## Abstract

Many primary immunodeficiencies (PIDs) are recognised as being associated with malignancies, particularly lymphoid malignancies, which represent the highest proportion of cancers occurring in conjunction with this underlying condition. When patients present with genetic errors of immunity, clinicians must often reflect on whether to manage antitumoral treatment conventionally or to take a more personalised approach, considering possible existing comorbidities and the underlying status of immunodeficiency. Recent advances in antitumoral immunotherapies, such as monoclonal antibodies, antigen-specific adoptive cell therapies or compounds with targeted effects, potentially offer significant opportunities for optimising treatment for those patients, especially with lymphoid malignancies. In cases involving PIDs, variable oncogenic mechanisms exist, and opportunities for antitumoral immunotherapies can be considered accordingly. In cases involving a DNA repair defect or genetic instability, monoclonal antibodies can be proposed instead of chemotherapy to avoid severe toxicity. Malignancies secondary to uncontrolled virus-driven proliferation or the loss of antitumoral immunosurveillance may benefit from antivirus cell therapies or allogeneic stem cell transplantation in order to restore the immune antitumoral caretaker function. A subset of PIDs is caused by gene defects affecting targetable signalling pathways directly involved in the oncogenic process, such as the constitutive activation of phosphoinositol 3-kinase/protein kinase B (PI3K/AKT) in activated phosphoinositide 3-kinase delta syndrome (APDS), which can be settled with PI3K/AKT inhibitors. Therefore, immunotherapy provides clinicians with interesting antitumoral therapeutic weapons to treat malignancies when there is an underlying PID.

## 1. Introduction

The increased incidence of cancer in patients suffering from an inherited immunodeficiency has been demonstrated by the data reported in several immunodeficiency registries [1,2]. The oncological event sometimes acts as the patient’s gateway to knowledge about the existence of their primary immunodeficiency (PID), and this underlying condition should then be formally characterised for treatment in parallel with the management of the malignancy. The overall reported incidence of cancer among patients with a PID varies from 4–25%, depending on databases [3,4], with an estimated relative risk of cancer ranging from 1.4–2.3 [1,2]. Incidence is significantly dependent on the nature of the deficiency since PIDs belonging to the family of DNA repair defects show an incidence of cancer up to 40% [4]. This incidence is more marginally increased in diseases with a predominance of immune dysregulations such as autoimmune lymphoproliferative syndrome (ALPS) or interleukin-10 receptor deficiency syndrome; it is intermediately higher with PIDs which predominantly involve antibody deficiencies, ranging from about 10% with common variable immunodeficiencies to about 25% with activated phosphoinositide 3-kinase delta syndrome (APDS) [4]. Moreover, it is interesting to note the distinctive distribution of cancer subtypes affecting patients with a PID. Indeed, lymphoid malignancies represent more than half of the cancers seen among subjects with a PID, followed by skin cancers and, to a lesser extent, solid tumors such as gastric cancers or tumors of the central nervous system [3,4,5]. Among lymphoid malignancies, non-Hodgkin lymphomas (NHLs) are more common than Hodgkin’s lymphomas and leukaemia, and diffuse large B cell lymphomas represent the largest subset of NHLs [3]. Considering the distribution of cancers observed in these inherited immune disorders, it is noteworthy that the underlying genetic defect may be variously involved in oncogenic processes. Some PIDs secondary to DNA repair disorders are strongly associated with the development of lymphoid cancers early in life; some PIDs caused by gene defects affecting immune cell homeostasis or antitumoral immunosurveillance function are associated with cancers developing later and caused by oncogenic mechanisms requiring serial cumulative second events [3]. However, whatever the subtype of malignancy and the underlying oncogenic mechanisms, managing cancer treatment with a concurrent PID is still looked upon as a challenge by most physicians [6]. Indeed, treatment options should be carefully considered according to the malignancy subtype and to pre-existing comorbidities due to the underlying disorder, which are present in a significant proportion of patients [7]. With regard to recent advances in antitumoral immunotherapies and their clinical development—especially in lymphoid malignancies—after briefly assessing the PIDs associated with cancer and exploring the oncogenic mechanisms involved, this review’s purpose is to highlight the possibilities of implementing these advances in the treatment of cancers with concomitant inherited immune disorders according to their underlying presumed oncogenic mechanisms.

## 2. Inherited Immune Disorders Associated with Cancer

A predisposition to cancer is a commonly reported characteristic of patients with many inherited genetic errors of the immune system [8,9]. These pairings are summarised in Table 1. The overall reported incidence of those cancers varies from 4–25% depending on the database, probably partially because of methodological issues but also, more interestingly, according to the subtypes of PID [4]. Indeed, when considering PIDs according to their underlying genetic defect, some are impressively associated with an increased risk of cancer whereas others are more marginally associated with malignant diseases [3]. In that respect, the highest incidence of cancer is seen among patients with the subset of PIDs due to DNA repair defects [10]. Thus, *DCLRE1C* (Artemis) deficiency and DNA ligase IV deficiency are radiosensitive, severe combined immunodeficiencies associated with a predisposition to malignancies [10,11], and ataxia-telangiectasia, Nijmegen breakage syndrome or Bloom syndrome are also typically characterised by a massive risk of cancer [12,13,14]. Ataxia-telangiectasia is caused by biallelic mutations in *ATM*, a gene involved in double-strand breaks in the DNA repair pathway and many other cellular pathways. The reported incidence of cancer in patients carrying *ATM* variants is about 20% [12]. Nijmegen breakage syndrome is an autosomal recessive syndrome provoked by hypomorphic mutations in *NBS1*. Nibrin, the encoded protein, forms a complex with RAD50 and MRE11, proteins involved in the repair of DNA double-strand breaks and the control of cell-cycle checkpoints. The reported incidence of malignancies in patients presenting with this condition is up to 40% [13]. Bloom syndrome is caused by biallelic variants in the *BLM* gene. The BLM protein belongs to the subfamily of RecQ helicase proteins and, via interaction with several other proteins, it acts as a gatekeeper for genome integrity. An incidence of cancer of about 50% is observed in patients carrying variants in the *BLM* gene [14].

The predominant malignancies associated with all these DNA repair defects are lymphoid malignancies. These include a large proportion of B-non Hodgkin lymphomas (B-NHLs), followed by acute lymphoblastic leukaemia and Hodgkin’s lymphoma [10]. However, interestingly, ataxia-telangiectasia, Nijmegen breakage syndrome and Bloom syndrome share the common feature of a varied distribution of malignancy subtypes depending on the patient’s age, with lymphoid malignancies predominating in the youngest patients and, with ageing, a progressively increasing incidence of solid tumors including carcinomas and brain tumors [10]. It is worth reflecting on the fact that lymphoid cells are under constant developmental stress to break, repair and mutate DNA during Ig/T-cell receptor rearrangements, and they would usually tolerate this without apoptosing. Hence, lymphoid cells are at a particular risk of not only suffering but also surviving, oncogenic mutations.

Some other PIDs which predispose patients to cancer are characterised by a defect in cellular pathways and this disrupts homeostasis in immune cells by arresting cell differentiation and maturation or by impairing apoptosis [3]. For example, in APDS, an autosomal dominant disorder, activating PIK3 delta mutations causes a maturation arrest of B cells and faster senescence of T cells. This results in a B cell lymphoproliferative syndrome with enlarged secondary lymphoid organs, which sometimes mimics lymphoma [15]. However, there is also the potential for malignant transformation with a resulting risk of B-cell non-Hodgkin lymphoma [16]. The impaired IL10 signalling pathway observed in IL10 receptor deficiency, the underlying pathophysiological cause of a proportion of very early onset inflammatory bowel diseases, as well as the STAT3 deficiency characterising Job syndrome and resulting from loss of function mutations in *STAT3*, are both involved in the malignant transformation of B lymphoid cells, presumably by compromising the physiological homeostasis of B-cell precursors [17,18]. In ALPS, the homeostasis of lymphoid cells is impaired by defects in B and T-cell apoptosis [19]. ALPS is caused by variants in many genes involved in apoptosis, particularly by an inherited variant in *TNFRSF6*. This gene encodes the Fas cell surface death receptor, a transmembrane protein belonging to the TNF-receptor superfamily promoting cell death signalling via a complex formed by the interaction between its cell-death domain, caspase 8 and caspase 10. ALPS patients bearing *TNFRSF6* variants are predisposed to various subsets of nonHodgkin and Hodgkin’s lymphomas [20].

Susceptibility to infection is one of the most evident features of inherited errors in the immune system. Defective immunoglobulin production and lymphocyte function disrupt adaptive immunity and so impair the control of host cells infected by latent viruses. The immune system’s importance in the control of cells infected with a latent virus is well known among immunocompromised patients following an organ transplant, when the reactivation of latent viruses such as the Epstein-Barr virus (EBV) can happen [21]. This condition occasionally triggers an uncontrolled proliferation of lymphoid cells, predisposing to a variety of lymphoproliferative disorders including lymphomas, and formally recognised as posttransplant lymphoproliferative disorders [22]. Similarly, a constellation of PIDs with underlying molecular mechanisms as varied as the gene variants of *WAS* in Wiskott-Aldrich syndrome, *RMRP* in cartilage-hair hypoplasia and *ITK* in interleukin-2-inducible T-cell kinase—or sometimes as yet undiscovered variants as in some common variable immunodeficiencies—predispose patients to malignant lymphoproliferative diseases linked to uncontrolled EBV disease [23].

## 3. Oncogenic Mechanisms Involved in PIDs Associated with Cancer

The mechanisms involved in oncogenesis are complex and multiple. However, it is noteworthy and intriguing that a scattering of PIDs predispose patients to malignancies with such a massive over-incidence of cancer that the question may be raised about whether there are shared pathways to immunodeficiency and oncogenesis. Interestingly, Hauck et al. presented a model reconciling the malignancies observed in conjunction with PIDs with their possible underlying oncogenic mechanisms [24]. Briefly, they showed that such intrinsic events as differentiation or apoptosis, cell signaling or DNA repair defects were sufficient per se to cause the early onset of myeloid or lymphoid malignancies. However, combinations of multiple other intrinsic events may provoke the extrinsic conditions prone to a later onset of malignancies where solid tumours predominate [24]. A reinterpretation of those features is proposed in Figure 1.

Lymphoid cell precursors are the sole somatic cells to physiologically compromise their genome stability as long as they proceed to V(D)J recombination, class-switch recombination and somatic hypermutation during their differentiation and maturation. V(D)J recombination is the recombination of DNA double-strand breaks which enable the rearrangement of heavy immunoglobulin (*IgH*) and *TCR* gene segments in B and T cell precursors, respectively, to produce diversity in the immune repertoire [25]. During this cellular process, lymphoid precursors are not exempt from containing mistakes arising during the V(D)J recombination steps, with a risk of translocation between the loci of *IgH* and *TCR* genes and the loci of genes specifically engaged in their corresponding stages of maturation [26]. Proteins such as Artemis, DNA ligase IV or nibrin are part of the nonhomologous end-joining complex—machinery involved in the repair of the DNA double-strand breaks generated during V(D)J recombination [26]. ATM serine/threonine kinase (ATM) and BLM protein are both involved in the machinery for repairing DNA double-strand breaks and in controlling cell-cycle checkpoints. Both proteins chaperone DNA double-strand break repair processes in lymphoid precursors and, more broadly, in any cells suffering from DNA damage [10]. Any defect in this machinery may jeopardise genomic integrity and exposes cells to primary oncogenic events such as somatic chromosome translocations, usually involving *IgH* or *TCR* loci [26,27]. These translocations are generally recognised as being powerful oncogenic events in animal models. In human models, this concept may be illustrated by the extremely high incidence of cancers in patients presenting with a constitutive DNA repair defect [24,27]. Indeed, these defects are characterised by the early onset of acute lymphoblastic leukaemia or lymphoma and the further development of secondary solid tumors [12,14,27].

In addition to errors in DNA repair, the molecular defects observed in some PIDs disrupt the homeostasis of lymphoid cells, which is more or less directly involved in the oncogenic process. Indeed, although many reports have demonstrated the role of PIK3/AKT/mTOR and NFkB pathways in lymphomagenesis [28,29], their constitutive activation, seen in APDS and IL10 receptor deficiency, respectively, and the defective apoptosis pathway characterising ALPS, result in the uncontrolled proliferation of lymphoid cells [16,17,19]. These conditions are prone to the development of cumulative secondary oncogenic events via an increased risk of unrepaired errors during immunoglobulin class-switch recombination and the somatic hypermutation processes. The consequence of this is a predisposition to malignant transformation in a proportion of patients presenting with those PIDs [3]. However, malignancies usually only happen once several secondary oncogenic errors have accumulated in the cell’s genome. Mature malignant B lymphoproliferations represent almost the only group of cancers observed in association with this subset of PIDs; unlike the case of DNA repair defects, they do not manifest themselves early in life but rather at different ages throughout it [3].

The immune system is mainly dedicated to infection control, with a significant role in clearing intrusive infectious agents and setting up immune memory or controlling latent viruses such as EBV, varicella-zoster virus or herpes simplex virus. EBV is a herpes virus affecting more than 90% of the population and characterised by the integration of its genome into the genome of B cells, resulting in a definitive latent persistence in these cells. Moreover, this integration demonstrates oncogenic properties [30]. Burkitt’s lymphoma—an endemic African disease linked to EBV infection—illustrates this feature. The immune system’s role in controlling EBV-infected B cells is obvious if we consider the pathophysiology of post-transplant lymphoproliferative disorders [31]. Along with an EBV reactivation disease, some transplant patients receiving immunosuppressive drugs develop a panel of lymphoid proliferations ranging from polymorphic lymphoid proliferation to highly aggressive B-cell lymphomas [22]. Similarly, some PIDs are associated with EBV-linked lymphoid malignancies, although in these cases, the mechanisms of oncogenic transformation are probably more complex than can be explained by EBV’s oncogenic power alone [23,32]. Indeed, only a minority of these patients present with a malignancy, and a secondary oncogenic event is probably required before the cancer appears [32]. In parallel with its function of controlling latent viruses, the immune system may have a role in anti-tumoral immunosurveillance through its earlier clearance of emerging malignant cells [33]; this concept is still the subject of debate. Thus, any inherited defect in immune function may compromise antitumoral immunosurveillance [33,34].

## 4. Possibilities of Immunotherapy for Children with Primary Immunodeficiency and Cancer

The over-incidence of cancers observed alongside primary and secondary immunodeficiencies demonstrates the immune system’s role in controlling the development of malignancy [3]. Consequently, the boom in antitumoral immunotherapy approaches seen in the past 20 years is unsurprising [35,36,37]. However, these approaches must cover quite a large field, ranging from the development of antitumoral vaccines to the use of engineered immune effectors redirected against malignant cells or, in some cases, the replacement of the defective immune system using allogeneic stem cell transplantation (allo-SCT) [38]. There are equally many approaches focusing on antitumoral immunotherapy in childhood patients presenting with an underlying PID—these are listed in Table 2. However, depending on the remaining immune function available in the patient’s body, which is required to enhance immunotherapy, each approach should be considered in relation to the nature of the patient’s inherited immune function impairment. Moreover, the complexity involved in treating immunocompromised patients with cancer arises, on the one hand, from an increased risk of the side effects of antitumoral treatments due to the frequently associated comorbidities linked to the underlying condition, and, on the other hand, from the risk of worsening immune defects, particularly when immunotherapies are applied [7]. Both of these aspects should be considered when choosing the appropriate therapy, and patients should be closely monitored throughout their treatment.

### 4.1. Humanised Monoclonal Antibodies

Antitumoral monoclonal antibodies probably represent the most widely explored means of delivering clinical immunotherapy [38,54]. Based on the successful use of the antiCD20 monoclonal antibody, rituximab, in the treatment of many CD20 + lymphoid malignancies [55,56], a collection of monoclonal antibodies targeting various epitopes present on malignant-cell surfaces has emerged [54,57]. Many are used in combination with conventional antineoplastic agents, and some have even demonstrated their efficacy against certain tumors as single agents [58]. Mechanistically, once they have coupled with their targets, monoclonal antibodies still require several mechanisms to trigger their antitumoral potency. Some antibodies can trigger apoptosis signaling just by binding to the target cells. In parallel with that direct antitumoral effect, the clearance of monoclonal antibodies binding to malignant cells may involve basic immune functions such as antibody-dependent cellular cytotoxicity (ADCC) and complement-dependent cytotoxicity (CDC) [59]. As well as their direct antiproliferative effects, both ADCC and CDC are involved in the antitumoral properties of rituximab. In an attempt to boost those effects and overcome resistance mechanisms, new generation antiCD20 antibodies were developed, some encouraging CDC (ofatumumab) [60] and some encouraging ADCC (obinutuzumab) [61]. Another application of monoclonal antibodies came from coupling them with antineoplastic drugs. The goal of these conjugated monoclonal antibodies is to restrict the delivery of the drug to the antibody-binding cells in order to increase its concentration in malignant cells and decrease its toxicity to healthy cells. Two conjugated monoclonal antibodies have been clinically successful against childhood cancer: antiCD33 coupled with ozogamycin to treat CD33 positive acute myeloid leukaemia [62] and antiCD30 coupled with auristatin E to treat anaplastic large-cell lymphoma and Hodgkin’s lymphoma [36,42].

The latest advance involving therapeutic monoclonal antibodies has been the development of bi-specific T-cell engager antibodies (BiTEs) [63]. These chimeric antibodies bear a double-valence—one steered towards an epitope on the malignant target cell surface and the other towards the surface T cell receptor (TCR) coreceptor CD3. The aim is to redirect antitumoral cytotoxic CD3 positive T cells to antibodies binding on malignant cells. Many phase II trials involving paediatric studies have demonstrated the significant benefits of using antiCD3 antiCD19 BiTE blinatumomab to treat CD19 positive preB cell acute lymphoblastic leukemia (preB ALL), especially in clearing up residual disease [45]. This treatment is therefore now being assessed as a front-line therapy in phase III trials for treating childhood preB ALL. Although, to the best of our knowledge, there is as yet no published data on using blinatumomab to treat preB ALL patients with PID, it could be an interesting approach to consider for patients requiring clearance of a residual disease before allo-HSCT. Indeed, blinatumomab has been successfully implemented in preB ALL paediatric patients as a bridge to transplant in cases of persistent minimal residual disease, suggesting that this therapy may be able to replace the additional courses of chemotherapy sometimes required by this condition. This approach may avoid the severe side-effects of chemotherapy which contribute to transplant-related mortality among patients suffering from DNA repair disorders. Residual T/NK cell cytotoxicity might be critical to the efficacy of blinatumomab, but it must not be impaired by a decreased number of T cells [64]. This residual disease-fighting function is not easy to measure accurately in the various subsets of PIDs, but in inherited immune defects such as APDS, STAT3 deficiency, X-linked lymphoproliferative syndrome type 1 or Wiskott-Aldrich syndrome, in which T/NK cell cytotoxicity is impaired [65,66], blinatumomab may be predicted to have a poor effect. In PIDs belonging to the DNA repair disorder subset, however, such as Nijmegen breakage syndrome or ataxia telangiectasia, which are characterised by a progressive decrease in T cell numbers but where a persistent residual function is expected, blinatumomab may be more beneficial.

Based on successful experiences with many monoclonal antibodies for the treatment of childhood cancers, implementing adoptive immunotherapy as a therapeutic strategy against cancers associated with a PID should be considered [7,39,42,58]. In cases of PID associated with DNA repair defects, in particular, monoclonal antibodies may replace, or at least allow a reduced chemotherapy dose intensity, thus preventing deleterious and sometimes life-threatening side-effects [39,58].

Nevertheless, the possible side-effects of using monoclonal antibodies should not be ignored. Some are the consequence of off-target effects, such as the neurotoxicity reported after taking blinatumomab or brentuximab [67], which may be an issue in cases where the central nervous system is primarily affected by a condition prone to neurodegenerative disorders, as it is in ataxia telangiectasia or Nijmegen breakage syndrome. Moreover, a cytokine release syndrome is observed in a proportion of patients receiving blinatumomab, and managing the most severe cases may require the administration of tocilizumab, an anti-interleukin 6 receptor antagonist [68]. This immunomodulator is itself not exempt from potential side-effects, especially in cases where there is already a primary immune defect [69]. Lastly, antiCD19/CD20 monoclonal antibodies cause a deep, long-lasting decrease in B cells. The subsequent hypogammaglobulinemia may worsen a pre-existing humoral defect. These parameters should be monitored in patients receiving these drugs, and substitutive immunoglobulin therapy may be required accordingly.

### 4.2. Cell Therapies

More evidence for the immune system’s caretaker role in cases of cancer is illustrated, once again, in post-transplant malignancies. Indeed, in post-transplant conditions, one important part of treatment is to partly restore the immune system by tapering immunosuppressive drugs [70]. It is noteworthy that this intervention may be sufficient to control some subtypes of post-transplant lymphoproliferative disorder. The most established cell therapy used to restore effective immunity in PIDs is allo-SCT [71,72]. Allo-SCT in PIDs in general, and in PID-related cancers in particular, requires special expertise, and its benefits should always be carefully weighed against its risks [73]. In cases of cancer with an underlying PID, allo-SCT may be considered with the twin aims of supporting intensified antitumoral treatment, if indicated, and of curing the primary immune defect prone to malignancy [7]. Nevertheless, allo-SCT is made all the more challenging by pre-existing comorbidities, the choice of an appropriate conditioning regimen, risk management for post-transplant infections, the choice of post-transplant immunomodulation and the need to monitor immune recovery [46]. Allo-SCT has been successfully implemented for cancers associated with PIDs, and the approach should definitely be considered for some of those conditions [7]. Special caution should be paid to PIDs caused by DNA repair disorders [47]. This constellation of disorders may require allo-SCT to correct immunodeficiency and to cure haematological malignancies. However, the risk of later onset of solid tumors characterising DNA repair disorder must be taken into consideration after allo-SCT [73]. This is due to the absence of any curative effect of allo-SCT in nonhematopoietic cells which retain a constitutive DNA repair defect and suffer from the conditioning regimen stress to DNA break. Therefore, because of this increased risk of early and late severe toxicities, conditioning regimens should be chosen accordingly. With the advent of reduced-intensity conditioning regimens, allo-SCT has demonstrated good results with PIDs related to DNA repair disorders, except for ataxia telangiectasia, where outcomes after allo-SCT remain dismal [47].

Although all the mechanisms involved in the antitumoral effects of allo-SCT have yet to be completely elucidated, there is strong evidence of an immune-mediated effect on residual malignant cells. This is known as the graft versus leukaemia (GvL) effect because it is typically observed in this subset of malignancies. Therefore, those observations provided the rationale for using more or less sophisticated antigen-nonspecific or antigen-specific adoptive cell therapies [38,74]. To magnify the GvL effect, using a donor lymphocyte infusion was proposed as an adjuvant to the allo-SCT, but with a significantly increased risk of GvHD. Then, in an attempt to mimic the GvL effect while avoiding GvHD, antigen-nonspecific adoptive cell therapy (ACT) approaches were developed. ACT increases the number of autologous or allogeneic cytotoxic T lymphocytes with antitumor specificity [74]. To date, successful clinical trials of this approach in childhood cancers remain scarce [75,76]. In some virus-mediated malignancies happening as a complication of a PID, some authors have proposed the use of specific antivirus cytotoxic T lymphocytes. Specific anti-EBV cytotoxic T-lymphocytes (CTLs) were developed. CTLs were also generated to target the EBNA1, LMP1 and LMP2 EBV proteins in EBV-mediated lymphomas, with significant clinical effect. This approach is not a gold standard in the front-line treatment of paediatric lymphomas, even in the event of EBV-mediated post-transplant lymphoproliferative disorders, but it could certainly be considered as a salvage therapy in cases of refractory EBV-mediated lymphoma in immunocompromised patients [21,77].

The most recent improvements in cell therapy have come with the development of chimeric antigen receptor T cells (CAR T cells) [49]. CAR T cells are genetically modified autologous T cells engineered to produce and carry a chimeric antigen receptor composed of a single-chain variable fragment with tumour cell antigen specificity, and mimicking variable heavy and light immunoglobulin domains. The chimeric antigen receptor is fused with the intracytoplasmic domain of CD3 combined with costimulatory domains. The receptor’s variable domain is then able to bind to a specific tumour cell antigen and trigger the cytotoxic effect of the T-cell carrier closely linked to the targeted tumoral cell. As a proof of concept, CAR T cells were first developed with anti-CD19 specificity, thereby targeting CD19 positive malignancies [78]. The most recent improvements in CAR T cell development mainly involve CD19/CD22 bi-specific CAR T cells [79]. The first clinical trials using CARs produced encouraging results in paediatric patients, and this cell therapy is now approved as a second-line treatment for relapsing/refractory CD19 positive preB ALL. In the near future, there is no doubt that the specificity of CARs will be extended to a larger panel of target antigens covering a broad subset of tumoral cells [35]. In cases involving malignancies with underlying inherited immunodeficiency, autologous CARs may be an interesting approach with which to substitute standard chemotherapy and avoid its toxicity. However, there are two major limitations to this. On the one hand, depending on the nature of the immune function affected, it may be questionable to use autologous immune T cells to generate CARs, with a risk of CAR T cells being functionally disturbed. On the other hand, in the event of an inherited immunity disorder, especially in the group of DNA repair defects, reinfusing genetically engineered autologous T lymphocytes might prove challenging. Both of these limitations might be overcome by the development of universal allogeneic CAR T cells derived from healthy donors [49].

### 4.3. Immunomodulators

Somatic mutations of the genes involved in the development of the immune system, such as *NOTCH1*, *IKZF1* or *JAK3*, are linked to oncogenic events in lymphoid malignancies. However, the constitutive alteration of genes involved in inherited immunodeficiencies associated with cancer are exceptionally oncogenic in themselves, and they may occasionally be involved in the polygenic mechanisms driving oncogenesis. Being able to explain the development of malignancies observed in association with PIDs by their underlying genetic defect, and consequently being able to treat that cancer accordingly by targeting the pathway involved using small molecules, is a very appealing approach. Unfortunately, to date, this has not been made real in clinical practice. Indeed, very few PIDs with close associations to cancer are due to defects in genes that are directly involved in targetable oncogenic pathways. One exception is APDS, a PID caused by gain-of-function mutations in *PIK3CD* [16]. The resulting constitutive activation of *PIK3CD* is responsible for initially nonmalignant, uncontrolled lymphoproliferation with consequently enlarged lymphoid structures [15]. Some APDS patients will subsequently develop malignant lymphoproliferation. Ongoing clinical trials are testing the benefits of using leniolisib, an inhibitor of PIK3, to control the lymphoproliferative disorder in APDS [51]. In the event of a malignant transformation, targeting the PIK3/AKT pathway with inhibitors of PIK3 might be considered in combination with conventional anticancer drugs, with the knowledge that this drug reportedly has limitations due to increased AID-mediated genomic instability [15,80]. Many gene defects causing PIDs highlight major key pathways involved in lymphoid cell development. Some can be targeted using small molecules, providing some authors with a rationale for using targeted immunotherapies in lymphoid malignancies [38]. For example, Bruton’s tyrosine kinase (BTK) signalling pathway is crucial for B cell differentiation and proliferation. The activity of BTK can be aborted by ibrutinib or acalabrutinib, which have both demonstrated their antitumoral effects on malignant B cells [53,81]. One clinical trial showed the feasibility of combining BTK inhibitors with chemotherapy in the difficult childhood condition of relapsing or refractory Burkitt’s lymphoma [52]. However, the resulting severe B-cell depletion and hypogammaglobulinemia provoked by those drugs must be considered if they are used as an adjuvant treatment to chemotherapy in lymphoid malignancies with an underlying PID. More importantly, Feldhahn et al. have suggested that BTK activity may be required to maintain the tumor-suppressant role of preB cell receptors in preB ALL [82]. Another example is the role of activated mTOR in immune cell proliferation and the downstream PI3K/AKT and mitogen-activated protein kinase kinase (MEK) signalling pathways. The mTOR inhibitors sirolimus and everolimus are immunosuppressive drugs occasionally used in solid organ transplantation and more marginally in haematopoietic stem cell transplantation. In addition to their immunosuppressive effects, many studies have strongly supported their antitumoral effects, and some research groups have strongly suggested switching any immunosuppressive treatment to mTOR inhibitors in the event of post-transplant lymphoproliferative disorders [83,84,85]. Activation of the PI3K/AKT/mTOR signalling pathway has been shown in ALPS, thus providing a rationale for targeting this pathway with mTOR inhibitors as this lymphoproliferative disease is associated with malignancy [86,87].

Over the last few years, several checkpoint inhibitors have been successfully developed in many cancer subsets [35]. They consist of monoclonal antibodies binding the surface receptors cytotoxic T-lymphocyte activated 4 (CTLA4) and programmed death 1 (PD-1), or its ligand programmed death-ligand 1 (PD-L1). CTLA4 is expressed on cytotoxic T cells, and PD-1 is expressed on various immune cells such as cytotoxic T cells, B cells, monocytes, dendritic cells and tumour-infiltrating lymphocytes. PD-L1 is present on the surface of tumoral cells. Interaction between these receptors and their ligands participates in the immune self-tolerance of tumours by disabling antitumoral T cell-mediated cytotoxicity [88]. The CTLA4 inhibitor ipilimumab, and the antiPD1s nivolumab and pembrolizumab, have been successfully clinically implemented against many cancers, particularly against lymphoid malignancies [50,89]. Attempting to emphasise antitumoral, CTL-mediated immune response, we could predict that this immunotherapeutic approach might be especially appropriate with those PIDs associated with a suspected loss of antitumoral immunosurveillance. However, once again, the effectiveness of this approach supposes that antitumoral CTL activity is preserved. Interestingly, combining checkpoint inhibitors with other immunotherapy approaches has also been gauged in clinical trials, and this may represent an attractive approach for enhancing the immune effectors required for adaptive and innate antitumoral immunity in patients presenting with constitutive underlying immune dysfunctions [88,90].

On the whole, although many small compounds targeting the immune system could be considered as candidates with antitumoral effects on malignancies associated with PIDs, there is, as yet, little clinical experience of their use and only a few have shown evidence of a strong enough rationale for their use.

## 5. Conclusions

The management of cancers in children with underlying primary immunodeficiencies is a clinical challenge requiring the greatest circumspection in terms of the accurate histological characterisation of the malignancies present, an understanding of the oncogenic mechanisms involved, appropriate treatment and adequate follow-up. Indeed, cases of primary immunodeficiency are sometimes associated with uncommon histological subsets of malignancies. Depending on the underlying genetic defect, the case may lead to increased toxicity to conventional treatment, an unexpected aggressiveness from the cancer or secondary malignancies. Based on their clinical experience, clinicians will often combine their analysis of these specificities to provide the patient with a very personalised treatment. Some immunotherapy approaches developed over the last few years in oncohaematology, and especially in lymphoid malignancies, the most frequent malignancies seen in conjunction with cases of PID, are now being approved, or are close to being approved, as front-line treatments. They offer great opportunities for improving treatment management for immunocompromised patients presenting with cancer. Some monoclonal antibody and CAR T cell approaches may therefore soon be in position to at least partly replace certain conventional chemotherapeutic agents, thus avoiding the frequent, severe side-effects occurring among sufferers of some PIDs. In some situations, improvements in allo-SCT will offer the opportunity to cure both the cancer and the underlying constitutive inherited error of immunity. Better knowledge about the molecular mechanisms involved in some PIDs, leading to the identification of the constitutively disturbed signalling pathways prone to the development of malignancies, may in the future provide clinicians with a rationale for targeted treatments. Both the need for residual T/NK cell antitumoral cytotoxicity for the effective initiation of T-cell therapies, and the risk of worsening the underlying primary immune defect, should be taken into account before considering antitumoral immunotherapy. Although there is no longer any doubt about the significant potential benefits of implementing some immunotherapies in the treatment of cancers associated with PIDs, one of the most important remaining worries is about how to clearly define a place for immunotherapy in each malignancy associated with each condition of PID. Unless larger proportions of lymphomas or leukaemia than have so far been reported are associated with underlying PIDs, considering randomised control trials to compare conventional treatments with immunotherapy will not be realistic because of the low number of patients suffering from such inherited disorders and diagnosed with cancer. A useful and feasible option for attempting to create more homogeneous therapeutic approaches to conditions encompassing immunotherapies could involve a common international database registering clinical experiences of the successful and unsuccessful treatments of any cancer with an associated PID.

## Figures and Tables

**Figure 1 biomolecules-10-01112-f001:**
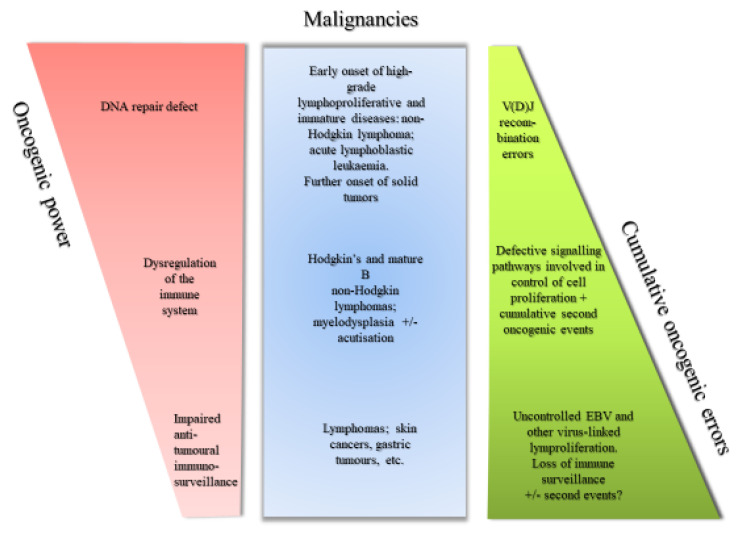
Illustrates oncogenic mechanisms of cancers with underlying PID.

**Table 1 biomolecules-10-01112-t001:** Lists primary immunodeficiencies (PIDs) prone to cancer and their corresponding gene defect.

	PID	Gene	Distinctive Features	Oncologic Phenotype	Supposed Mechanism
Chromosome breakage syndromes	Radio-sensitive SCID				
	DCLRE1C (Artemis) deficiency	DCLRE1C	T-B-NK + radio-sensitive SCID, decreased Ig Hypomorphic mutants: hypo IgG, lymphopenia, Omenn syndrome	EBV positive B cell lymphoma	Defect in V(D)J and class-switch recombination
	DNA ligase IV deficiency	LIG4	T-B-NK + radio-sensitive SCID, decreased Ig, microcephaly, Omenn syndrome, pancytopenia	EBV positive B cell lymphoma; leukemia	Defect in V(D)J and class-switch recombination
	Ligase I deficiency	LIG1	Decreased T cells, normal B cells, low IgA and IgGGrowth retardation	Lymphoma	Defect in class-switch recombination
	Nijmegen breakage syndrome	NBS1	Progressive decreased T cells, reduced B cells, low IgA, IgE and IgG subclasses, increased IgM; microcephaly, dysmorphism	Lymphoma, solid tumors	Chromosome instability, defect in V(D)J and class switch recombination, defect in somatic hypermutations
	Bloom syndrome	BLM	Normal T and B cells, reduced production of IgG; short stature, dysmorphism, sun-sensitivity; bone marrow failure	Leukemia, lymphoma	Chromosome instability, Defect in class-switch recombination
	Ataxia teliangiectasia	ATM	Progressive decreased T cells, normal B cells, low IgA, IgE and IgG subclasses, increased IgM; ataxia, telangiectasia	Leukemia, lymphoma, solid tumors	Chromosome instability, Defect in V(D)J and class-switch recombination
	PMS2 deficiency	PMS2	Normal T cells, low B cells, low IgG and IgA, increased IgM; café-au-lait spots	Leukemia, lymphoma, brain tumors, colorectal carcinoma	Defect in class-switch recombination and somatic hypermutations
	MCM4 deficiency	MCM4	Normal T and B cells, low NK cells, normal Ig; Short stature	Lymphoma	Chromosome instability
Dysregulation of the immune-system	SCID ADA deficiency	ADA	Severe combined immunodeficiency with low T cells, B cells and NK cells, low Ig	Lymphoma	Immune dysregulation +/− uncontrolled EBV-linked lymphoproliferation
	Autoimmune lymphoproliferative syndrome FAS	TNFRSF6	Increased TCR ab double negative T cells, low memory B cells; splenomegaly, adenopathies; autoimmune cytopenias	Lymphoma	Defect in lymphocyte apoptosis
	APDS	PIK3CD	Decreased CD4 T cells with reversed CD4/CD8 ratio, decreased B cells, low IgG and IgA, high IgM	Lymphoma	Constitutive activation of PIK3 may act downstream BCR/CD19 by promoting B cell proliferation via AKT/mTOR
	IL10 receptor deficiency	IL10-Ra IL10-Rb	Normal T and B cells; leukocytes fail to respond to IL10 cytokine	Lymphoma	Constitutive activation of NFkB pathway; loss of immunosurveillance (?)
	STAT3 deficiency (Job syndrome)	STAT3	Normal total T and B cells; decreased unswitch and switch memory B cells; hyper IgE, decreased specific antibodies; facial dysmorphism; bone fragility	Lymphoma	Defective antitumoral immunosurveillance
Loss of the immuno-control of infections	CVID not overwise specified	Unknown	Hypo IgG and IgA +/− IgM	Lymphoma, skin cancer, gastric cancer	Uncontrolled infectious agent-linked lymphoproliferation (?) +/− loss of immunosurveillance (?)
	Cartilage-hair hypoplasia	RMRP	From normal to variably decreased T Cells, normal B cells, normal or reduced Ig; short limb dwarfism; bone marrow failure	Lymphoma	Uncontrolled EBV-linked lymphoproliferation
	X-linked lymphoproliferative syndrome type 1	SH2D1A	Normal or increased activated T cells, low memory B cells; HLH features triggered by EBV infection	Lymphoma	Defective antitumoral immunosurveillance; uncontrolled EBV-linked lymphoproliferation
	CD27 deficiency	CD27	Normal T cells, absence of memory B cells, reduced Ig; HLH features triggered by EBV infection; bone marrow failure	Lymphoma	Uncontrolled EBV-linked lymphoproliferation
	CTPS1 deficiency	CTPS1	Normal or decreased T and B cells, increased IgG	Lymphoma	Uncontrolled EBV-linked lymphoproliferation
	RASGRP1 deficiency	RASGRP1	Normal number of T and B cells, increased IgA	Lymphoma	Uncontrolled EBV-linked lymphoproliferation
	CD70 deficiency	CD70	Low Treg, normal B cells; reduced IgG, IgA and IgM	Lymphoma	Uncontrolled EBV-linked lymphoproliferation
	ITK deficiency	ITK	Progressive decreased T cells, normal B cells, normal or low Ig.	Lymphoma	Uncontrolled EBV-linked lymphoproliferation
	XMEN	MAGT1	Low CD4 T cells and recent thymic emigrant cells, normal B cells, normal Ig	Lymphoma	Uncontrolled EBV-linked lymphoproliferation
	Wiskott Aldrich syndrome	WAS	Progressive decreased T cells, normal B cells; low IgM, high IgA and IgE.	Lymphoma	Uncontrolled EBV-linked lymphoproliferation
	WHIM syndrome	CXCR4	Decreased B cells, hypogammaglobulinemia, neutropenia. warts	Lymphoma	Uncontrolled EBV-linked lymphoproliferation
	EVER1 deficiency	TMC6	Predisposition to human papillomavirus infection	Skin cancer	Uncontrolled HPV infection
	EVER2 deficiency	TMC8	Predisposition to human papillomavirus infection	Skin cancer	Uncontrolled HPV infection
PID with myelodysplasia	Dyskeratosis congenita	Many genes involved	Decreased T cells, variably decreased B cells, variable hypogammaglobulinemia; short telomeres; bone marrow failure; abnormality of skin, hair and nails	Myelodysplasic syndrome Solid tumors	Genetic instability,
	Congenital neutropenia				
	Elastase deficiency	ELANE	Neutropenia	Myelodysplasic syndrome/Leukemia	Genetic instability, Cumulative second mutational genetic events (CSF3R, RUNX1) and chromosomal aberrations (monosomy 7, trisomy 21)
	Kostmann disease	HAX1	Neutropenia; neurological symptoms (developmental delay, epileptic seizures)	Myelodysplasic syndrome/Leukemia	
	Shwachman-Diamond syndrome	SBDS	Neutropenia; exocrine pancreatic insufficiency	Myelodysplasic syndrome/Leukemia	

SCID: severe combined immunodeficiency

**Table 2 biomolecules-10-01112-t002:** Summarises possibilities of immunotherapy in cancer associated with PIDs.

Malignancy Subset	Immunotherapy	Oncogenic Mechanisms			References
		DNA Repair Defect	Dysregulation of Immune System	Loss of Virus and Antitumoral Immune Control	
- CD20 + lymphoid malignancies - CD22 + lymphoid malignancies - Hodgkin’s lymphoma/ALCL - CD33 + myeloid malignancies - CD19 + lymphoid malignancies	**Monoclonal antibodies** Monospecific antibodies - AntiCD20: Rituximab Obinutuzumab - AntiCD22: Epratuzumab Conjugated antibodies - AntiCD30 + auristatin E: Brentuximab vedotin - AntiCD33 + ozogamycin: Mylotarg Bispecific antibodies (BiTE) - CD19/CD3 BiTE: Blinatumomab	+++ + + +++ ++ ++ ^(*)^	+++ + + +++ ++ ++ ^(*)^	+++ + + +++ ++ ++ ^(*)^	[7,39] [40] [41] [42] [43] [44,45]
- Any haematological malignancy if indication exists - EBV-driven lymphoid malignancies - CD19 + lymphoid malignancies - Any malignancy expressing the targeted surface marker	**Cell therapies** - Allo-HSCT - antiEBV CTL - autologous chimeric-antigen receptor T cells (CARs T cell): Targeting CD19 Targeting any further markers expressed on tumoral cell surface	+++ ^(°) (#)^ - ++ (?) ^(*) (§)^ + (?) ^(*) (§)^	+++ +++ ++ (?) ^(*) (§)^ + (?) ^(*) (§)^	+++ +++ ++ (?) ^(*) (§)^ + (?) ^(*) (§)^	[7,46,47] [7,38] [48] [35,49]
Multiple cancers: - Solid tumors, including melanoma, central nervous system tumors, neuroblastoma - Haematological malignancies including Hodgkin’s and nonHodgkin lymphomas, acute leukemias - Lymphoid malignancies with constitutive PI3K/AKT/mTOR pathway activation - Mature B cell malignancies engaging B cell receptor signalling	**Immunomodulators** Checkpoint inhibitors - cytotoxic T-lymphocytes-associated protein 4 (CTLA4) inhibitors: Ipilimumab - programmed death receptor 1 (PD-1): Nivolumab Pembrolizumab - programmed death ligand receptor 1 (PD-L1): Atezolizumab - mTOR inhibitors: Sirolimus - PIK3/AKT inhibitors: Leniolisib - BTK inhibitors: Ibrutinib Acalabrutinib	+ ^(*)^ + ^(*)^ + ^(*)^ + ^(*)^ - - - -	+ ^(*)^ + ^(*)^ + ^(*)^ + ^(*)^ +++ (ALPS) +++ (APDS) + -	+ ^(*)^ + ^(*)^ + ^(*)^ + ^(*)^ - - - -	[35,50] [35,50] [35,50] [35,50] [19] [51] [52,53]

+++ immunotherapy supported by published data in primary immunodeficiencies. ++ immunotherapy for which published data are missing in primary immunodeficiencies but supported by encouraging published clinical data in nonimmunocompromised paediatric patients. + immunotherapy supported by published preclinical or clinical data providing a proof of concept. – no supportive data to our knowledge. ^(*)^ conditioned by persistent T cell cytotoxicity; ^(°)^ caution with choice of conditioning regimen; ^(#)^ with the exception of Ataxia telangiectasia. ^(§)^ question of gene transfection of lymphoid cells bearing constitutive genetic defect to be addressed particularly in cases involving a DNA repair defect. (?) questionable mainly because both limitations mentioned above.

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
