# Peer review of "The Possibilities of Immunotherapy for Children with Primary Immunodeficiencies Associated with Cancers"

_biomolecules, 2020, doi:10.3390/biom10081112_

Round 1

Reviewer 1 Report

Major points

The manuscript requires additional input to improve the English language.  There is a significant number of grammatical errors and on several occasions the wrong word is chosen.  I have listed many of these in my minor points below, but not all I’m afraid.  This element must be improved to allow clear reading and interpretation of the manuscript. There are also several sentences which have been stopped and separated by a paragraph.

The description of the variable association between different PID and the malignancies which have been reported to date is interesting.  I appreciate this is not the focus of the manuscript, but it clearly sets the scene for the subsequent discussion of potential therapies.  The author highlights that the nature of any one specific PID diagnosis impacts on probable oncogenic mechanisms and I think the description linking that to the clinical experience, eg DNARD have intrinsic DNA damage risk high incidence of malignancy early in life… is nice.  However, alternative schema to the one proposed in Figure 1 have previously been presented, eg Hauck/Voss/Seidel JACI 2018, but these are not acknowledged.  The central issue here is that the nature of the immunodeficiency impacts on the risk/nature of the malignancy, and (critical for this manuscript) also impacts on the choice of therapy.  This is mentioned but does not have a sufficiently central position in the discussion.

One example of the importance of this link is the use of BiTE antibodies, which is not adequately discussed here, either in Table 2 or in the text.  They require endogenous T lymphocyte activity to function yet seem to be strongly recommended in all three PID categories in Table 2, without further discussion of the implication of individual disease types and residual immune function. This is also discussed in lines 216-226, but without reference to any published example of them being used in PID.  There is some suggestion that reference 47 (line 226) supports the use of Blina in PID, but this is simply a case report of 2 children who received rituximab.  Furthermore, it would also be wrong to assume that just because these therapies are targeted they are free from potential toxicity.  Indeed, understanding the impact of recipient immuno-dysregulation on toxicity must be key if suggesting a drug such as Blinatumumab which carries notable and life-threatening side effect of cytokine release syndrome. Equally, the impact of the well recognised neurotoxicity of drugs such as Blinatumumab, but also eg Brentuximab, in patients with neurodevelopmental or neurodegenerative conditions such as AT/NBS must be carefully considered.

The formatting of Table 2 makes it hard to follow which entry relates to which therapy as they are not in line. Most importantly however, it is not clear what the +, ++, +++ nomenclature denotes in the third-fifth columns.  I assume it relates to the clinical utility of that approach, but this is not stated and it is not clear on what basis this scoring is given. This is critical as this table gives the suggestion that specific therapies have an established role in particular subgroups of PID (denoted by +++).  To suggest this for any drug other than rituximab is misleading and risks patients being exposed to unmonitored toxicity on a case-by-case basis.  The meaning of the scoring system should have been explicitly stated.

The discussion around allogeneic transplant is superficial and does not consider published data on toxicity, transplant related mortality or outcome (eg Cohen et al, Blood 2007, Prunotto et al, JACI 2020). The discussion moves straight on the value of DLI, then CTLs and rapidly into CAR-T cells.  There is no discussion about the complexity of allogeneic transplant in this patient group, no discussion of the different complexities and role for HSCT in DNARD versus other PID, see multiple publications by ESID Inborn errors working party.  This is the most established immunotherapy for this patient group and requires a much more detailed discussion.

Small molecule inhibitors may well offer promise in this area and the example of PI3Kd inhibition is well received.  However, BTK inhibition is far from established in paediatric B cell malignancy in which NFkB signalling plays no major role. The given reference relating to its combination with chemotherapy for relapsed refractory B-NHL (ref 59) relates to toxicity and tolerability, not efficacy. The statement in line 305 that the use of sirolimus is “robustly documented” is not supported by any references meaning the reader cannot assess the robustness for themselves. This section is far too speculative but without acknowledging it.

Minor points

Abstract – “Face to patients…”  This sentence does not make sense to me.  Do you mean “Faced with patients…”?

Line 28 – should read “suffering”

Line 74 – “Reparation” should read “repair”

Line 81-82 – The sentence is incomplete.  It appears that it has been separated from the second half of the sentence, now on line 85 beneath table 1.

Line 88-90 – It is worth reflecting on the fact that lymphoid cells are under a constant developmental stress to break, repair and mutate DNA during Ig/receptor rearrangements and would usually tolerate this without apoptosing.  Hence they are at particular risk of not only suffering, but also surviving oncogenic mutations.

Line 130/131 – there should not be a break in the sentence

Line 136 – “reparation” should read “repair”

Line 139 – “chaperons” should, I think, read “chaperone”

The sentence beginning on line 173, “Making a parallel…” doesn’t make sense.

The sentence beginning on line 179, “With regard to…” doesn’t make sense.

Table 2 – Mylotarg, not Milotarg

Line 202 - there should not be a break in the sentence

Author Response

Reviewer #1

We have really appreciated the high-level expertise of the reviewer and we are grateful for the input provided by his comments to improve the manuscript. All have been carefully taken into consideration and addressed.  

Major points

The manuscript requires additional input to improve the English language.  There is a significant number of grammatical errors and on several occasions the wrong word is chosen.  I have listed many of these in my minor points below, but not all I’m afraid.  This element must be improved to allow clear reading and interpretation of the manuscript.

The revised version of the manuscript has been sent to a professional native speaker English editor before resubmission.

There are also several sentences which have been stopped and separated by a paragraph.

This point was mainly due to the reformatting process of the manuscript and has been corrected in the current version.  

The description of the variable association between different PID and the malignancies which have been reported to date is interesting.  I appreciate this is not the focus of the manuscript, but it clearly sets the scene for the subsequent discussion of potential therapies.  The author highlights that the nature of any one specific PID diagnosis impacts on probable oncogenic mechanisms and I think the description linking that to the clinical experience, eg DNARD have intrinsic DNA damage risk high incidence of malignancy early in life… is nice.  However, alternative schema to the one proposed in Figure 1 have previously been presented, eg Hauck/Voss/Seidel JACI 2018, but these are not acknowledged. 

Figure 1 was originally drawn based on our knowledge and our understanding of oncogenic mechanisms involved in malignancies frequently met in PIDs. However, we agree the reference cited above and previously published nicely support our own interpretation of those oncogenic mechanisms. Then, we have extensively referred to this article to support our purpose as mentioned lines 138-144. We propose Figure 1 as a reinterpretation of these published features.       

The central issue here is that the nature of the immunodeficiency impacts on the risk/nature of the malignancy, and (critical for this manuscript) also impacts on the choice of therapy.  This is mentioned but does not have a sufficiently central position in the discussion.

Indeed, that is one of the major goal of our message in this work. We have generally highlighted that critical point lines 59-61; lines 218-224; lines 435-437 to try to make it stronger. We have addressed that point further in details along the following comments.    

One example of the importance of this link is the use of BiTE antibodies, which is not adequately discussed here, either in Table 2 or in the text.  They require endogenous T lymphocyte activity to function yet seem to be strongly recommended in all three PID categories in Table 2, without further discussion of the implication of individual disease types and residual immune function. This is also discussed in lines 216-226, but without reference to any published example of them being used in PID. 

That point has been addressed lines 258-273

There is some suggestion that reference 47 (line 226) supports the use of Blina in PID, but this is simply a case report of 2 children who received rituximab. 

Apologise. That point was due to a formatting issue now corrected line 276. The reference referred to the experience with monoclonal antibodies.

Furthermore, it would also be wrong to assume that just because these therapies are targeted they are free from potential toxicity.  Indeed, understanding the impact of recipient immuno-dysregulation on toxicity must be key if suggesting a drug such as Blinatumumab which carries notable and life-threatening side effect of cytokine release syndrome. Equally, the impact of the well recognised neurotoxicity of drugs such as Blinatumumab, but also eg Brentuximab, in patients with neurodevelopmental or neurodegenerative conditions such as AT/NBS must be carefully considered.

Both points has been emphasised lines 279-299

The formatting of Table 2 makes it hard to follow which entry relates to which therapy as they are not in line. Most importantly however, it is not clear what the +, ++, +++ nomenclature denotes in the third-fifth columns.  I assume it relates to the clinical utility of that approach, but this is not stated and it is not clear on what basis this scoring is given. This is critical as this table gives the suggestion that specific therapies have an established role in particular subgroups of PID (denoted by +++).  To suggest this for any drug other than rituximab is misleading and risks patients being exposed to unmonitored toxicity on a case-by-case basis.  The meaning of the scoring system should have been explicitly stated.

Table 2 has been reformatted to make it more readable. Columns 1 and 2 have been inverted. Scoring meaning has been explained in the legend and corresponding references added. 

The discussion around allogeneic transplant is superficial and does not consider published data on toxicity, transplant related mortality or outcome (eg Cohen et al, Blood 2007, Prunotto et al, JACI 2020). The discussion moves straight on the value of DLI, then CTLs and rapidly into CAR-T cells.  There is no discussion about the complexity of allogeneic transplant in this patient group, no discussion of the different complexities and role for HSCT in DNARD versus other PID, see multiple publications by ESID Inborn errors working party.  This is the most established immunotherapy for this patient group and requires a much more detailed discussion.

A more extensive discussion on HSCT and its challenging points in PID has been added as well as additional references. Please see lines 296-315.

Small molecule inhibitors may well offer promise in this area and the example of PI3Kd inhibition is well received.  However, BTK inhibition is far from established in paediatric B cell malignancy in which NFkB signalling plays no major role. The given reference relating to its combination with chemotherapy for relapsed refractory B-NHL (ref 59) relates to toxicity and tolerability, not efficacy.

BTK inhibition was an example to illustrate some attempts in using compounds targeting pathways involved in immune system. We agree that is yet very preliminary and with some limitations. This was clarified lines 380-384

The statement in line 305 that the use of sirolimus is “robustly documented” is not supported by any references meaning the reader cannot assess the robustness for themselves.

Some references added line 391.

This section is far too speculative but without acknowledging it.

This has been acknowledged lines 411-414.

Minor points

Abstract – “Face to patients…”  This sentence does not make sense to me.  Do you mean “Faced with patients…”? : done

Line 28 – should read “suffering”: done

Line 74 – “Reparation” should read “repair”: done

Line 81-82 – The sentence is incomplete.  It appears that it has been separated from the second half of the sentence, now on line 85 beneath table 1.

Formatting issue, apologise for that. Corrected

Line 88-90 – It is worth reflecting on the fact that lymphoid cells are under a constant developmental stress to break, repair and mutate DNA during Ig/receptor rearrangements and would usually tolerate this without apoptosing.  Hence they are at particular risk of not only suffering, but also surviving oncogenic mutations.

We thought the reviewer suggested to insert that sentence, so we did. Lines 96-99.

Line 130/131 – there should not be a break in the sentence

Formatting issue, apologise for that. Corrected

Line 136 – “reparation” should read “repair”: done

Line 139 – “chaperons” should, I think, read “chaperone”: done

The sentence beginning on line 173, “Making a parallel…” doesn’t make sense. Rewritten

The sentence beginning on line 179, “With regard to…” doesn’t make sense. Rewrittten

Table 2 – Mylotarg, not Milotarg: done

Line 202 - there should not be a break in the sentence. Rewritten

Reviewer 2 Report

This is a good review of malignancy in Primary Immunodeficiencies including the challenges of treating these patients in a conventional way, and the possibility of using various immunotherapeutic options.

The main problem is that it needs to be edited by a native English speaker as sometimes the sense is lost due to grammatical errors. 

In addition greater attention needs to be paid to long-term cure of these patients as merely treating their malignancy will still leave them with their underlying PID and susceptibility to malignancy. The following reference may help with this.

Prunotto G, Offor UT, Samarasinghe S, Wynn R, Vora A, Carpenter B, Gowdy C, McHugh K, Windebank KP, Rovelli AM, Slatter MA, Gennery AR, Veys P, Bacon CM, Bomken S, Lucchini G. HSCT provides effective treatment for lymphoproliferative disorders in children with primary immunodeficiency.

J Allergy Clin Immunol. 2020 May 1:S0091-6749(20)30626-6. doi: 10.1016/j.jaci.2020.03.043. Online ahead of print.

Author Response

Reviewer #2

This is a good review of malignancy in Primary Immunodeficiencies including the challenges of treating these patients in a conventional way, and the possibility of using various immunotherapeutic options.

We have appreciated being encouraged by a reviewer expert in the field.  

The main problem is that it needs to be edited by a native English speaker as sometimes the sense is lost due to grammatical errors. 

The revised version of the manuscript has been sent to a professional native speaker English editor.

In addition greater attention needs to be paid to long-term cure of these patients as merely treating their malignancy will still leave them with their underlying PID and susceptibility to malignancy. The following reference may help with this.

That critical point has been taken into consideration. Lines 307-311. The suggested reference has been embedded.

Prunotto G, Offor UT, Samarasinghe S, Wynn R, Vora A, Carpenter B, Gowdy C, McHugh K, Windebank KP, Rovelli AM, Slatter MA, Gennery AR, Veys P, Bacon CM, Bomken S, Lucchini G. HSCT provides effective treatment for lymphoproliferative disorders in children with primary immunodeficiency.

J Allergy Clin Immunol. 2020 May 1:S0091-6749(20)30626-6. doi: 10.1016/j.jaci.2020.03.043. Online ahead of print.

Round 2

Reviewer 1 Report

The amended manuscript presents a substantially more balanced review of the field.  The references are more comprehensive and the English writing is now greatly improved.

Reviewer 2 Report

Thanks for the amendments.